# Non-Destructive Detection of Meat Quality Based on Multiple Spectral Dimension Reduction Methods by Near-Infrared Spectroscopy

**DOI:** 10.3390/foods12020300

**Published:** 2023-01-08

**Authors:** Xiaochun Zheng, Li Chen, Xin Li, Dequan Zhang

**Affiliations:** Institute of Food Science and Technology, Chinese Academy of Agriculture Sciences, Key Laboratory of Agro-Products Quality and Safety Control in Storage and Transport Process, Ministry of Agriculture and Rural Affairs, Beijing 100193, China

**Keywords:** non-destructive detection, meat quality, dimension reduction, near-infrared spectroscopy

## Abstract

The potential of four dimension reduction methods for near-infrared spectroscopy was investigated, in terms of predicting the protein, fat, and moisture contents in lamb meat. With visible/near-infrared spectroscopy at 400–1050 nm and 900–1700 nm, respectively, calibration models using partial least squares regression (PLSR) or multiple linear regression (MLR) between spectra and quality parameters were established and compared. The MLR prediction models for all three quality parameters based on the wavelengths selected by stepwise regression achieved the best results in the spectral region of 400–1050 nm. As for the spectral region of 900–1700 nm, the PLSR prediction model based on the raw spectra or high-correlation spectra achieved better results. The results of this study indicate that sampling interval shortening and of peak-to-trough jump features are worthy of further study, due to their great potential in explaining the quality parameters.

## 1. Introduction

Meat is very important for people, as it provides protein, fat, and vitamins to the body. The consideration of the public regarding meat quality has gradually increased in recent years. Meat quality is typically evaluated according to many properties, such as its color, tenderness, water-holding capacity, pH, moisture content, protein content, and fat content, which can be determined through instrumental laboratory analyses or artificial sensory evaluation [1]. These methods are reliable and classical, but also costly and time-consuming, therefore being unable to obtain real-time results and/or to be used for online detection [2]. Near-infrared spectroscopy (NIRS) refers to light in the range from 750 nm to 2500 nm, which reflects the information of hydrogen groups of chemical substances. Hence, it can be used to predict chemical composition information, and therefore further utilized to predict the quality parameters or authenticity of meat and meat products [3,4,5,6].

As a high-potential technology, near-infrared spectroscopy (NIRS) has been widely studied and used for non-destructive detection of meat quality in recent decades, due to its advantages of being non-invasive, labor-free, and time-saving [7,8,9]. However, the strong motivation for near-infrared spectroscopy to be developed rapidly and applied commercial is closely related to the miniaturization of NIR spectrometers and the progress of chemometrics [10]. A great number of commercial applications of NIRS have been reported, ranging from the advancement of high-volume fabrication of microelectromechanical systems (MEMS) to micro-mirror arrays [11]. At the same time, a variety of benchtop/transportable/portable/hand-held/smartphone-based NIR devices have been developed, in line with advances in chip and computer technology [12,13]. NIR devices have presented huge application prospects, with their continuously smaller size and cheaper price. However, the development of the meat industry has put forward higher requirements for both the miniaturization and stability of NIR non-destructive testing devices.

Fresh meat is a complex detection object as it contains thousands of chemical substances, leading to complexity in its near-infrared optical characteristics [14]. This means that we may not be able to obtain the optical characteristics of a Gaussian-type peak, as is typically obvious in liquid chromatography or mass spectrometry. Therefore, a variety of spectral feature extraction methods or algorithms have emerged, according to relevant requirements [15,16,17,18,19,20,21]. Furthermore, meat quality prediction models usually cannot be improved through complicated and advanced chemometrics treatment algorithms. The resolution of near-infrared spectrometers is becoming higher and higher with their development in the past decades [22], but the response values of adjacent wavelengths in the same kind of samples are still highly correlated. So, whether a high resolution is necessary, considering the complex composition of the fresh meat due to the overlapping features and highly repetitive information, remains an open question. On the other hand, wavelength selection is another popular method to obtain a stable model by developing many complex or simple algorithms, such as Fisher’s linear discriminant transformation, regression coefficients and variable importance in projection (VIP), competitive adaptive re-weighted sampling (CARS), Monte Carlo uninformative variables elimination (MCUVE), genetic algorithms (GAs), and so on [23,24,25,26,27,28,29]. There have been many different studies focused on different objects, achieving good or bad performances based on the use of NIR technology.

The most common method is to build a prediction model using the whole spectrum, composed of hundreds or even thousands of wavelengths. This not only makes the modeling vulnerable to high interference of redundant information, but this also leads to high costs and a large volume of spectral devices. Wavelength selection can be considered a good approach to simplify the workload and to increase the efficiency; however, raw meat is an extremely complex bio-based material, its near-infrared spectrum reflects a lot of complex information of the material, and the relevant chemical bond characteristics are not obvious. Thus, scholars have devoted a lot of research energy to developing new wavelength screening algorithms, which still may not be efficient or, even able to meet the requirements.

Therefore, four different dimension reduction methods were tested in this study, including stepwise regression, extraction of high correlation spectra, sampling interval shortening, and peak-to-trough jump features. These methods were utilized to build predictive models for the contents of protein, fat, and moisture in lamb meat. The specific objectives of this research were: (1) To develop two new methods to extract spectral characteristics from visible-near-infrared spectra; (2) to compare the prediction performance of calibration models with different sources of spectral data, including raw spectra and four different methods of spectral feature extraction; and (3) to establish predictive models for meat quality based on the use of simple, high-potential methods to extract the spectral features.

## 2. Materials and Methods

### 2.1. Sample Preparation

All Small-Tailed Han lamb meat samples used in this study were collected from 25 lamb carcasses randomly selected from the production line of a slaughter hall in Inner Mongolia. The lambs were raised on local farm for 7–9 months. The experiment lasted for 5 days, which means that five carcasses were chosen per day. These carcasses were chilled for 24 h at a temperature of 4 °C immediately in a pre-cooling storage with other carcasses after slaughter, and then pulled out onto the segmentation line. The topside, backstrap, oyster, fillet, thick flank, and tender loin were removed from the left side of each carcass for collection of spectra and determination of all quality parameters.

### 2.2. Collection of NIR Spectra

In this study, visible/near-infrared spectra were considered, in the range 400–1700 nm, including two wavebands 400–1050 nm and 950–1700 nm, were collected to test the research hypotheses. A self-developed near-infrared spectroscopy system (ST1) was utilized to collect the spectra in the range 400–1050 nm, which mainly consisted of four parts: A light source (built using a 20 W halogen lamp with a reflective cup, bracket for holding, distance module, and power supply unit), a spectrometer (AvaSpec-ULS2048CL-EVO-RS, Avantes Inc., Apeldoorn, The Netherlands), a portable collection probe (composed of pick-up optical fiber in the middle and transmitting optical fiber around), and a self-developed collection software (based on Visual Studio 2017, C++). Furthermore, another spectrometer (45 × 42 mm, Micro NIR^TM^, JDSU, Breinigsville, PA, USA) was a pocket device applied to collect the spectra in the 950–1700 nm region.

These two NIR spectral systems were connected to a laptop by USB cables and preheated for 30 min before collection. All the cuts were transferred to the laboratory of the slaughterhouse immediately after removal from the carcasses, and the visible fat was cut away to reduce adverse effects. It took about 30 min to get samples ready for collection of spectra. Black and white references were acquired at the beginning of acquisition for both spectral systems. The sample was placed on a black plate, the spectra from 200–1100 nm of the samples were obtained by placing the portable probe on the surface of the sample softly (exposure time, 100 ms; three scans were taken and spaced evenly for each sample and five scans were collected from the sensors to calculate average spectra; the sampling interval, 0.56 nm; 1607 data points in total). The spectral information collected by the pick-up fiber was transferred to the operational computer and converted to digital signals. The spectra from 900–1700 nm of the samples were obtained in a similar manner (exposure time, 250 ms; three scans were taken and spaced evenly for each sample and five scans were collected from the sensors to calculate average spectra; sampling interval, 6.25 nm; 125 data points in total). The samples were frozen as soon as possible and transported back to the laboratory of the Chinese Academy of Agricultural Sciences (CAAS) in Beijing under the cold chain to determine the quality index of the meat.

### 2.3. Analysis of Meat Quality

The protein content, crude fat content, and moisture content of the samples were determined according to the national standards of China. All samples were first homogenized using a mixer before measuring meat quality.

The protein content was tested according to the following procedure [30]: (1) A 1 g sample was weighed accurately (0.001 g) and transferred to a digestion tube. Then, 0.4 g copper sulfate, 6.0 g potassium sulfate, and 10 mL concentrated sulfuric acid were added into the tube; (2) the sample was digested to transparent by putting the digestion tubes into a digester, and continued for 1 h after the temperature of the digester had reached 420 °C; (3) the digestion tube was moved to a Kjeldahl nitrogen analyzer for liquid addition, distillation, and titration automatically after cooling to room temperature, and the test results were automatically calculated by the analyzer (the factor of nitrogen to protein was set to 6.25).

The fat content was tested according to the following procedure [31]: (1) A 2 g sample was weighed accurately (0.001 g) and transferred into a piece of filter paper cartridge; (2) the cartridge was put into the extraction cylinder of a Soxhlet extractor, and the receiving bottle was then connected after drying to constant weight and added with anhydrous petroleum ether (with boiling range at 60–90 °C) to 2/3 of the solvent; (3) then, the extraction process was completed by boiling (30 min), reflux (about 90 min), and solvent recovery (about 30 min) in turn, where the temperature of the heater was set to 130 °C; (4) the receiving bottle was moved to an oven at a temperature of 100 ± 1 °C until it reached constant weight (defined as the difference between two weights less than 2 mg, the weighing frequency was 1 h with 30 min cooling time in a dryer), and the weight of the receiving bottle was recorded; (5) the fat content was calculated as follows: *F_C_* = (*m*_1_ − *m*_0_)/*m* × 100%, where *m*_1_ is the total weight of the fat in the sample and the receiving bottle, *m*_0_ is the weight of receiving bottle, and *m* is the weight of the sample.

The moisture content was tested according to the following procedure [32]: (1) An aluminum box to be used for drying, containing 15 g sea sand and a small glass rod, was placed in an oven and dried to constant weight (defined as the difference between two weights being less than 2 mg, the weighing frequency was 1 h with 30 min cooling time in a dryer) at a temperature of 103 ± 2 °C; (2) 5 g (accurate to 0.001 g) of ground sample was weighed and removed to the aluminum box, and 10 mL ethanol (95%) was added and mixed with the sample and the sand. The ethanol was evaporated to dryness in a water bath at a temperature of 70 ± 1 °C; (3) the aluminum box and its contents was removed to an oven at a temperature at 103 ± 2 °C and dried to a constant weight (defined as the difference between two weights being less than 1 mg, the weighing frequency was 1 h with 30 min cooling time in a dryer); (4) the moisture content was calculated as follows: *W_C_* = (*w*_2_ − *w*_3_)/(*w*_2_ − *w*_1_) × 100%, where *w*_1_ is the total weight of the aluminum box, glass rod, and sand; and *w*_2_ and *w*_3_ are the total weights of the aluminum box, glass rod, sand, and sample before and after drying, respectively. 

### 2.4. Wavelength Selection

In this study, the spectral data of each sample included hundreds, or even thousands, of spectral data points (125 and 1607), which may contain redundant data that would introduce noise. Thus, the selection of optimal wavelengths is very essential for extracting the most efficient data and speeding the analysis process up. A group of methods are available for selecting important wavelengths [33]. As a classic method, stepwise regression (SWR) was used to eliminate redundant variables, in order to choose more effective variables and obtain a more stable model. The variable information selected by SWR is representative, but it can easily lose information through removing relevant information, such that it is necessary to compare SWR with other methods, in order to evaluate its applicability.

### 2.5. Characteristic Extraction

There is no doubt that it is wise to use feature extraction or wavelength selection algorithms to extract spectral characteristics [34]. However, some simple methods are also occasionally effective. Therefore, four kinds of spectral information, including full spectra, selected wavelengths, highly correlated spectra, and jump information (as described in Section 2.5.3), were used to build the prediction models for lamb meat quality parameters, and the results were analyzed and compared. Their descriptions are provided in the following sections.

#### 2.5.1. Extraction of Highly Correlated Spectra

Spectra related to quality parameters with high correlation, which we called highly correlated spectra (HCS), were chosen to build a correlation model for comparison with the model using the full spectra. For example, by choosing the reflectance values of 150 samples at 400 nm and calculating the correlation coefficients between the reflectance values and the protein content values of the 150 samples, the correlation coefficient at the 400 nm band, R_400_, was obtained. In this way, the correlation coefficient could be obtained between every reflectivity value and each of the quality index values for each wavelength. Using three quality indices, we obtained three different vectors composed of correlation coefficients.

#### 2.5.2. Reducing of Sampling Interval

The spectral data of 400–1050 nm bands were chosen in the manner of 1 out of 2, 1 out of 3, 1 out of 4, 1 out of 10, 1 out of 12, 1 out of 20, 1 out of 30, …, 1 out of 100, in order to obtain reduced spectral sets, which resulted in 25 spectral sets, including 565, 376, 282, 113, 94, 56, …, 13 data points, respectively. The spectra data at 900–1700 nm were processed similarly, choosing 1 out of 2, 1 out of 3, …, 1 out of 24. In this way, we obtained 22 data sets, including 62, 41, …, 5 data points, respectively.

#### 2.5.3. Extraction of Peak-to-Trough Jump Features

It was found that the jump features (difference of reflectance) from peak-to-trough (JFPT) also provide obvious features, through the observation of the raw spectra, which were considered worthy of further study. The jump features were calculated by subtracting the reflectance value at the trough from that at the adjacent peak, until all peaks and troughs had been traversed. The significance of this method is that the macro change in the reflectivity can reflect the change in chemical components, such as protein, fat, and moisture content, similar to the methods of high-performance liquid chromatography or mass spectrometry, which could carry out quantification according to the peak height or peak area.

### 2.6. Data Preprocessing

In order to obtain an accurate and efficient model for detection of protein, fat, and moisture content in lamb meat using visible/near-infrared spectroscopy, some typical spectral preprocessing methods, such as standard normalize transformation (SNV), multiple scatter correction (MSC), and Savitzky–Golay smoothing (S-G), were all applied to improve the accuracy of the prediction model. Among these methods, SNV was used to correct errors caused by scattering, MSC was used to remove the drift from baseline and enhance the spectral specificity, while S-G is a basic method for noise reduction.

### 2.7. Model Development

Multiple linear regression (MLR) is the most basic method in NIR quantitative analysis, and is also one of the most common modeling methods. It is a mathematical tool that quantifies the relationship between a dependent variable and one or more independent variables. Partial least squares regression (PLSR) is a linear and supervised multivariate calibration method, which is particularly suitable when the matrix of predictors has more variables than observations, and when there is a high correlation among the raw predictors. Therefore, these two mathematical approaches were adopted as modeling methods for comparison and specific uses. The correlation coefficient (R_C_ and R_P_) and root mean square error (SEC and SEP) of the calibration set and prediction set were calculated, respectively, and used to evaluate the advantages of the various models and methods. These indicators are common and direct parameters for evaluating models. In particularly, calibration models with higher R_C_ and R_P_ and lower SEC and SEP were identified as better models, and R_P_ should be smaller than R_C_ in general.

### 2.8. Software and Program

For this study, the visual studio software (2017 edition, Microsoft Corporation, Washington, DC, USA) was applied to develop an operation software for the self-developed NIR system. The pre-treatment of spectral data, the selection of variables, and the construction of the models were all implemented by using the MATLAB software (2014 edition, MathWorks Corporation, Natick, MA, USA).

## 3. Results and Discussions

### 3.1. Results of Meat Quality and Raw Spectra

The results of protein content, fat content, and moisture content are summarized in Table 1. The content of protein ranged from 14.85% to 23.02%, with an average content of 19.72%, which is basically consistent with values previously reported in the literatures [4,35]. The content of fat ranged from 0.55% to 16.13%, with an average content of 3.55%, resulting in a wide range, which can improve model construction. The content of moisture ranged from 64.94% to 78.70%, with an average content of 75.03%, which also fell within the wide range of typical results for fresh meat. The contents of protein, fat, and moisture depend on a variety of factors, such as farm, breed, sex, age, weight, and different cuts. Except for the chosen breed (Small-Tailed Han Sheep), all other facts were treated as a black box and not considered when designing the experiment, in order to obtain more abundant samples. As shown in Table 1, all three quality parameters obtained quite expected results, due to the advantage of random sampling in the production line and fusion of six different cuts of meat.

In this research, raw spectra ranged from 200 nm to 1100 nm (1607 data points) and 900 nm to 1700 nm (125 data points) were collected after correction by a black and white reference board, respectively. The spectral region with high noise caused by low response intensity was cut off. Therefore, the effective spectral region ranging from 400 nm to 1050 nm. A total of 1130 data points were obtained and analyzed for the first spectral region, and the reflective spectra of both spectral regions were corrected and calculated according to white and black references. The raw reflective spectra of all samples are shown in Figure 1A,B, respectively. It can be seen, from Figure 1, that the different spectral regularity of change was indicated in the same spectral region (900–1050 nm) from different NIR systems, largely due to the difference in type of light source or detector. The appearance and trend of the spectra showed obvious consistency in this study, and in general, were similar to those in other reports [34,36]. As shown in Figure 1, the troughs near 415 nm, 540 nm, 580 nm, 760 nm, 980 nm, 1180 nm, and 1450 nm in the two wavelength regions indicated where the absorption bands characterized the myoglobin and oxymyoglobin content. The peaks and troughs at 734 nm, 759 nm, 820 nm, 880 nm, 960 nm, 980 nm, 1080 nm, 1180 nm, and 1450 nm were related to third overtone of O–H (759 nm) and C–H (960 nm), second overtone of O–H (980 nm) and N–H (1080 nm), second overtone of C–H (1180 nm), and first overtone of O–H (1450 nm) and N–H (1450 nm), respectively [10,37,38]. All these spectral characteristics contributed to the construction of the quality parameter prediction models. 

### 3.2. PLSR Models Based on Full Spectra 

The PLSR models based on the full spectra of both wavelength regions for predicting the content of protein, fat, and moisture were implemented. The calibration model between raw spectra and pre-treated spectra to the different quality indices were built, and then applied to the validation set. The statistical results for both the calibration and validation sets are summarized in Table 2. As shown in Table 2, for the wavelength region of 400–1050 nm, the models for moisture content achieved better results in the model for protein content and fat content, the *R_C_* and *R_P_* were similar (0.88 and 0.84, respectively) as those based on SNV treatment, with lower standard errors. The prediction model obtained poor performance for moisture content based on raw spectra, for which the *R_P_* was only 0.53. The better models for fat were achieved with spectra after being treated by SNV or MSC, with *R_C_* and *R_P_* of 0.85 and 0.80, respectively. The best model for protein content was obtained with spectra processed with MSC with higher *R_P_* of 0.78. As for the wavelength region of 900–1700 nm, the prediction model for fat showed the best results (with *R_C_* and *R_P_* of 0.85 and 0.76, respectively) when based on spectra pre-treated by MSC or SNV. The modeling results for protein and moisture content were improved obviously by the pre-treatment methods of MSC and SNV, respectively, with *R_P_* of 0.74 and 0.75, respectively.

The modeling results for meat quality were basically consistent with the literature [10], where the modeling results for the 400–1050 nm spectral region were better than those for the spectral region of 900–1700 nm, according to the obtained results. However, these results were lower than those of some models constructed based on minced/ground lamb meat [34]. 

### 3.3. Modeling Results Based on Selected Spectra and Extracted Features

As described in Section 3.1, the spectra of these two wavelength regions included 1130 and 125 data points, respectively. However, data spectra with high resolution are not always necessary for modeling, some collinearity data were clearly observed, especially for the spectral region of 400–1050 nm, with high resolution. Therefore, the spectra collected from the meat samples probably contained information correlated with the target as well as wavelength regions with poor correlations [33]. A common way to select effective bands or extract effective features is through the use of certain algorithms [37], including (a) the use of highly correlated spectral regions for both of the two sets of spectra, which were selected and then used to build PLSR models for quality parameters; (b) selecting characteristic wavelengths of different quality parameters by the SWR algorithm, and then constructing prediction models; (c) applying sampling interval shortening to select spectral features, in order to construct the prediction models; and (d) using the JFPT to build the prediction models. 

#### 3.3.1. PLSR Models Based on High-Correlation Spectral Region 

In order to observe the correlation between the raw spectral data and the quality parameters, the coefficients between the reflectance of each band and the physicochemical indices were calculated. Meanwhile, multivariate scattering correction (MSC) was performed on the reflectance spectra, in order to reduce the physical effect of scattering. In this way, the coefficient for each wavelength was calculated, as plotted in Figure 2. The change rules of the three chemical components were quite similar in the spectral region of 400–1050 nm (Figure 2A), in which the spectral region of 600–850 nm presented a trend of positive correlation with fat content and a negative correlation with moisture content and protein content; while the trend was opposite in the next spectral region (850–1050 nm, Figure 2A). More than half of the absolute correlation coefficients were higher than 0.30, with the absolute correlation coefficients between reflectance and moisture content mostly ranging from 0.3 to 0.5.

The spectral region of 900–1700 nm (Figure 2B) showed less obvious regularity in different spectral regions; however, it was obvious that the change pattern of the correlation coefficients of fat content was inverse to that of moisture content and protein content, consistent with the case in 400–1050 nm. The correlation coefficients for fat content and moisture content were stronger than those for protein, with nearly half of the absolute correlation coefficients greater than 0.20, while the absolute correlation coefficients between reflectance and protein content mostly ranged from 0.1 to 0.2. It is well-known that the characteristic absorption of chemical bonds (C–H, N–H, O–H, S–H) is mainly concentrated in the wavelengths of 415 nm, 540 nm, 580 nm, 760 nm, 980 nm, 1180 nm, 1450 nm, and 1650 nm in the spectral region of 400–1700 nm, which could be observed clearly near these characteristic absorption wavelengths from the displayed correlation coefficient between the quality parameters and reflectance value of wavelengths. The correlations between single wavelengths and quality parameters implied that a lot of redundant information may be contained in near-infrared spectral data. Prediction models should be built and compared, in order to test whether highly correlated spectral data could provide good or more representative modeling results. 

For the 400–1050 nm spectral region, bands with correlation coefficients greater than 0.3, 0.4, and 0.5 were extracted for fat content, protein content, and moisture contents, respectively. For the 900–1700 nm spectral region, the bands with correlation coefficients greater than 0.1, 0.2, and 0.3 were extracted, respectively. The prediction models for corresponding indices were constructed for fat content, moisture content, and moisture content based on the extracted high-correlation spectral data, and the statistics of the modeling results are summarized in Table 3. It can be seen, from Table 3, that most of the model results were better than those based on full spectra. For the wavelength region of 400–1050 nm, the modeling results of protein content, fat content, and moisture content achieved better *R_P_* values (0.80, 0.81, and 0.84, respectively). For the wavelength region of 900–1700 nm, the modeling result based on highly correlated spectra showed a slight decrease for protein content and fat content but achieved a better result for moisture content than protein content and fat content, with the *R_P_* of the prediction models being 0.69, 0.75, and 0.84, respectively. The above results indicate that the high-correlation spectral region could represent the spectral information effectively, allowing us to build an effective quality prediction model. However, it is obvious that, regardless of whether the modeling results are based on the full spectra or high-correlation spectral region, the correlation coefficients of the prediction set for each of the quality parameters are quite low, which is not very satisfactory. Some possible reasons are as follows: (1) the spectral data and values of quality indices in this study came from a mixed sample set containing six different kinds of meat cuts, the consistency of the spectra was poor; (2) in general, the ampler the quality indices, the better the modeling results can be improved, especially when covering a wider range; (3) fat content changed more in this study, and the modeling results were better for fat than moisture content or protein content. 

#### 3.3.2. MLR Model Results Based on Selected Characteristic Wavelengths

Stepwise regression is one of the most common and typical algorithms for the selection of characteristic wavelengths. The characteristic wavelengths selected by SWR in the two spectral regions are summarized in Table 4, and MLR modeling results are presented in Table 5. As shown in Table 4, the characteristic wavelengths of different quality parameters differed: there were 8, 6, and 26 characteristic bands selected for protein, fat, and moisture, respectively, for the spectral region of 400–1050 nm; and 3, 14, and 6 characteristic bands selected for the spectral region of 900–1700 nm. Many particularly similar wavelengths were screened out, as can be observed from the table, such as 962 nm, 963 nm, and 967 nm for protein content and 951 nm, 952 nm, and 953 nm for moisture content in the spectral region of 400–1050 nm. The same situation occurred in the spectral region of 900–1700 nm, such as 1032 nm, 1057 nm, and 1069 nm for fat content. Some commonly important wavelengths were selected by looking through all the characteristic wavelengths, such as those wavelengths near 960 nm and 1212 nm. Encouraging model results were obtained, when based on the spectral region of 400–1050 nm, and the modeling results for protein, fat, and moisture content were improved to varying degrees. The improvement in the modeling result for moisture content increased the most obviously, with *R_P_* increasing from 0.84 to 0.90, while the correlation coefficients of protein and fat models increased from 0.78 and 0.80 to 0.79 and 0.83, respectively. For the spectral region of 900–1700 nm, the modeling results decreased slightly, and the correlation coefficients of the prediction set for three parameters were reduced to 0.55, 0.73, and 0.70 for the contents of protein, fat, and moisture, respectively. However, it still could be shown that the selected wavelengths represented most of the effective information. The modeling results for each quality index and the characteristic wavelength data were compared and analyzed, with the reason that the moisture content was better potentially being due to more wavelengths being selected.

#### 3.3.3. PLSR Modeling Results Based on Sampling Interval Shortening Method 

A simple method for increasing spectral sampling interval was also implemented in this study. According to the sampling interval shortening method, 25 sets of spectral data were obtained for the spectral range of 400–1050 nm, and 22 data sets were obtained for the spectral range of 900–1700 nm. After different pre-treatments, the PLSR model for fat content was constructed, and the statistics of these model results are presented in Figure 3. 

As shown in Figure 3A–C, for the spectral region of 400–1050 nm, the modeling results remained stable in a certain range with the increase in sampling interval. The PLSR model for predicting the fat content built using that spectral set including 125 bands (the sampling interval was 5.2 nm) showed better performance, with *R_P_* value close to 0.80 with or without pre-treatment. When the sampling interval continued to increase, the modeling results deteriorated sharply. Using the smallest spectral set (13 data points), the modeling results worsened, with the *R_C_* and *R_P_* both decreasing to 0.75, regardless of whether pre-treatment was utilized or not. 

As shown in Figure 3D–F, for the spectral region of 900–1700 nm, the modeling results remained relatively stable in a certain range with the process of sampling interval shortening, and decreased rapidly after a certain node, which is similar to the results for the 400–1050 nm spectral region. The spectral set including 25 bands (with sampling interval of 30.7 nm) achieved the best prediction of fat content, with *R_P_* close to 0.80 after pre-treatment by SNV and MSC. When the sampling interval continued to increase, the modeling results deteriorated sharply. Using the smallest spectral set (5 data points), the modeling results decreased, with the *R_P_* ultimately less than 0.5.

The results indicate that spectra with sampling interval exceeding 5.2 nm for the 400–1050 nm spectral region and 30.7 nm for the 900–1700 nm region may be unable to explain the chemical composition of the fresh lamb meat samples. The *R_C_* and *R_P_* values did not appear to significantly increase with higher spectral resolution, even when building the models using all variables. This could reflect the general rule that a higher resolution may not fully contribute to improvement of the modeling results after reaching a critical point. The above results provide some important advice for the development and application of small equipment: the most appropriate resolution and sampling interval should be determined for certain indices of certain objects, following which the minimum resolution and sampling interval that can satisfy each quality index may be confirmed, which is potentially very helpful in developing practical equipment with reduced cost.

#### 3.3.4. PLSR Modeling Results Based on Peak Jumping Feature

The peaks and troughs in the spectra could be clearly observed from the raw spectra in the 400–1050 nm region (Figure 1), which are marked clearly in Figure 4A. A total of four peaks and five troughs were obtained, and the spectral features appear similar to mountain slopes (Figure 4B); as such, we called them jump features from peak-to-trough (JFPT). Considering that the positions of the peak and trough bands were distributed evenly—that is, each trough sandwiched between two peaks, and similarly there are two troughs nearby every peak—a total of eight JFPTs were extracted from raw spectra by calculating the reflectance differences between peaks and adjacent troughs. As shown in Figure 4C, the changes (JFPT) from peak point to trough point appeared sharply different along with different samples, which can be seen in each of the sub-figures (Figure 4A–C). 

The MLR prediction models for fat content, protein content, and moisture content were implemented (Table 5), respectively, in order to verify whether this spectral feature could explain the quality information. Although the modeling results were not particularly satisfactory, when compared to the PLSR models based on pre-treated spectra or selected wavelengths, the JFPT could still represent most of the spectral information as some of the results were close to modeling results of SWR, especially in spectral region of 900–1700 nm, and obtained better results compared to raw spectra in Table 2. Therefore, the JFPT is worthy of further consideration in the field of non-destructive detection of meat quality based on NIR. 

## 4. Conclusions

In the current study, two new simple methods, including sampling interval shortening and peak-to-trough jump features, were explored for the extraction of spectral characteristics. These are easy to implement methods, where the former is more mechanical but can reduce the equipment volume, while the latter selects variables in a manner similar to stepwise selection. In this way, along with stepwise selection and extraction of highly correlated spectra, four methods were used and compared to extract spectral characteristics. For the spectral range of 400–1050 nm, stepwise selection still achieved the best results for all three chemical parameters of lamb meat, compared to the other three methods, while the model built using the raw spectral data performed better compared to data pre-treated with SNV and MSC. However, the other three feature extraction methods still performed quite well, with only slightly lower modeling results. The *R_P_* values of the optimal prediction model for moisture content, fat content, and protein content were 0.80, 0.81, and 0.84, respectively. This means that, regardless of whether the prediction model was constructed using a complex algorithm to select characteristic wavelengths or a simple method of dimension reduction to select wavelengths, similar modeling results could be achieved, proving that high-cost near infrared spectrometer devices with high resolution may not be necessary. The purpose of reducing data redundancy could be achieved by simple manual reduction of sensitivity, through developing a device with only 200 data points (instead of 1607) in the spectral region of 400–1050 nm or 25 data points (instead of 1607) in the spectral region of 900–1700 nm, which is conducive to reducing the cost and testing time for meat detection, as well as helping to develop smaller portable NIR devices. In other words, there was no large difference between the larger and smaller spectrometers when the predictive abilities are not overly different, the smaller one will be more welcomed for its size and low cost, thus providing theoretical support for the development of online, portable, and even micro-detection equipment.

## Figures and Tables

**Figure 1 foods-12-00300-f001:**
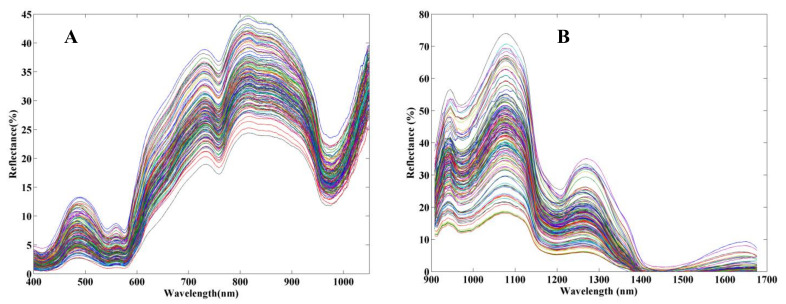
The raw reflective spectra of the samples in two regions. (**A**) refers to Vis/NIR spectroscopy ranging from 400 to 1050 nm of the sample, (**B**) refers to the spectrum of samples which ranged from 950 to 1700 nm, spectral curves of different color come from different samples.

**Figure 2 foods-12-00300-f002:**
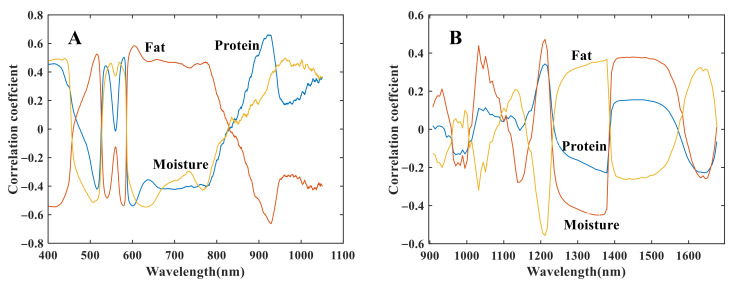
Correlation coefficients between raw spectral data and measured values of fat content, protein content, and moisture content: (**A**) showed the correlation coefficients between data in 400–1050 nm region and measured value of quality parameters; (**B**) showed the correlation coefficients between data in 900–1700 nm region and measured value of quality parameters.

**Figure 3 foods-12-00300-f003:**
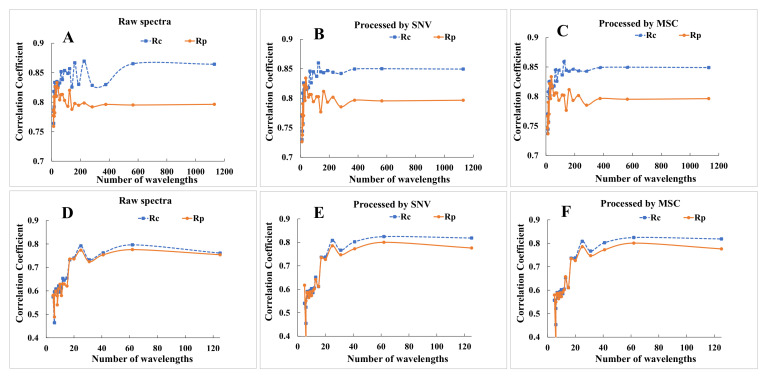
Modeling result of fat after sampling interval shortening and pretreating with SNV and MSC, (**A**–**C**) are the modeling results based on the spectral region of 400–1050 nm after increasing sampling interval and different pre-treatment methods; (**D**–**F**) are the modeling results based on the spectral region of 900–1700 nm after increasing sampling interval and different pre-treatment methods.

**Figure 4 foods-12-00300-f004:**
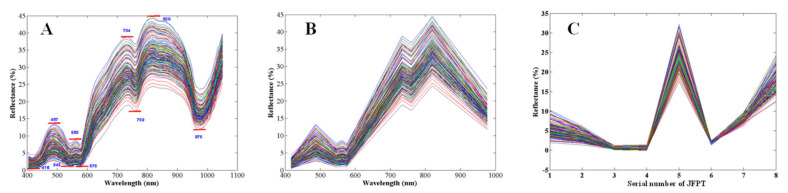
Peak and troughs featured in the spectral range of 400–1050 nm. (**A**), Peaks and troughs in raw spectral data; (**B**), virtual spectrum composed of peaks and troughs; (**C**), peak-trough jump features formed by the difference between peaks and troughs, spectral curves of different color come from different samples.

**Table 1 foods-12-00300-t001:** Determination of results of moisture content, protein content, and fat content.

Quality	Average	Max	Min	Range	Coefficient of Variation (%)
Protein (%)	19.72	23.02	14.85	8.17	6.61
Fat (%)	3.55	16.13	0.55	15.57	88.42
Moisture (%)	75.03	78.70	64.94	13.77	3.72

**Table 2 foods-12-00300-t002:** PLSR modeling results based on raw reflective spectra.

Spectral Region	Quality	Pre-Treatment	*R_C_*	*SEC*	*R_P_*	*SEP*
400–1050 nm	Protein	Raw	0.72	0.87	0.65	1.07
		SNV	0.86	0.61	0.77	0.84
		S-G	0.71	0.88	0.66	1.06
		MSC	**0.84**	**0.65**	**0.78**	**0.85**
	Fat	Raw	0.86	1.44	0.80	2.03
		SNV	**0.85**	**1.58**	**0.80**	**1.83**
		S-G	0.86	1.49	0.80	2.02
		MSC	**0.85**	**1.58**	**0.80**	**1.83**
	Moisture	Raw	0.62	2.30	0.53	3.06
		SNV	**0.88**	**1.27**	**0.84**	**1.47**
		S-G	0.61	2.36	0.53	3.07
		MSC	0.88	1.28	0.83	1.48
900–1700 nm	Protein	Raw	0.31	2.01	0.51	2.42
		SNV	0.78	0.82	0.74	0.89
		S-G	0.28	2.29	0.32	2.25
		MSC	**0.78**	**0.82**	**0.74**	**0.88**
	Fat	Raw	0.76	2.09	0.75	2.01
		SNV	**0.85**	**1.72**	**0.76**	**1.93**
		S-G	0.75	2.13	0.76	1.97
		MSC	**0.85**	**1.72**	**0.76**	**1.93**
	Moisture	Raw	0.15	7.10	0.23	9.71
		SNV	**0.78**	**1.80**	**0.75**	**1.80**
		S-G	0.17	7.77	0.13	8.80
		MSC	0.78	1.79	0.75	1.81

*R_C_*—correlation coefficient of calibration set, *R_P_*—correlation coefficient of prediction set, *SEC*—root mean square error of calibration set, *SEP*—root mean square error of prediction set; Bold refers to the model selected due to possessing the lowest *R_P_* value.

**Table 3 foods-12-00300-t003:** PLSR modeling results based on spectral data with high correlation between meat quality and spectra.

Spectral Region	Quality	Threshold of Correlation Coefficient	Number of Variables	*R_C_*	*SEC*	*R_P_*	*SEP*
400–1050 nm	Protein	0.3	828	**0.80**	**0.73**	**0.80**	**0.80**
		0.4	558	0.80	0.72	0.79	0.79
		0.5	100	0.52	1.05	0.52	1.11
	Fat	0.3	880	0.81	1.75	0.81	1.75
		0.4	661	**0.83**	**1.67**	**0.81**	**1.72**
		0.5	217	0.78	1.85	0.75	2.00
	Moisture	0.3	716	**0.87**	**1.33**	**0.84**	**1.53**
		0.4	441	0.84	1.42	0.81	1.67
		0.5	118	0.81	1.56	0.78	1.76
900–1700 nm	Protein	0.1	78	**0.77**	**0.84**	**0.69**	**0.93**
		0.2	22	0.63	1.02	0.62	0.98
		0.3	3	0.36	1.31	0.37	1.19
	Fat	0.1	98	**0.76**	**2.10**	**0.75**	**2.00**
		0.2	70	0.80	1.92	0.73	2.04
		0.3	31	0.78	1.99	0.71	2.11
	Moisture	0.1	105	0.78	1.78	0.72	1.91
		0.2	**80**	**0.78**	**1.76**	**0.82**	**1.54**
		0.3	57	0.74	1.93	0.78	1.68

*R_C_*—correlation coefficient of calibration set, *R_P_*—correlation coefficient of prediction set, *SEC*—root mean square error of calibration set, *SEP*—root mean square error of prediction set; Bold refers to the model selected due to possessing the lowest *R_P_* value.

**Table 4 foods-12-00300-t004:** Characteristic wavelengths of quality parameters based on stepwise regression algorithm.

Spectral Range	Quality	Selected Wavelengths (nm)
400–1050 nm	Protein	853, 858, 897, 942, 962, 963, 967, 986
	Fat	443, 492, 804, 805, 886, 926
	Moisture	441, 444, 490, 491, 493, 559, 562, 568, 600, 601, 607, 622, 722, 886, 933, 934, 937, 940, 951, 952, 953, 961, 969, 984, 996, 997
900–1700 nm	Protein	914, 1212, 1435
	Fat	914, 964, 1032, 1057, 1069, 1113, 1156, 1175, 1199, 1212, 1267, 1342, 1360, 1670
	Moisture	914, 964, 1199, 1212, 1342, 1360

**Table 5 foods-12-00300-t005:** Multiple linear regression modeling results based on the selected wavelengths and peak valley jump characteristics without pre-treatment.

Data Source	Spectral Region	Quality	*Rc*	*R_P_*
Selected wavelengths by stepwise regression	400–1050 nm	Protein	0.80	0.79
		Fat	0.82	0.83
		Moisture	0.94	0.90
	900–1700 nm	Protein	0.55	0.55
		Fat	0.80	0.73
		Moisture	0.66	0.70
Jump features from peak-to-trough	400–1050 nm	Protein	0.72	0.66
		Fat	0.71	0.74
		Moisture	0.70	0.70
	900–1700 nm	Protein	0.52	0.59
		Fat	0.66	0.73
		Moisture	0.63	0.72

## Data Availability

Data are available from the corresponding author upon request.

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
