# Peer review of "Non-Destructive Detection of Meat Quality Based on Multiple Spectral Dimension Reduction Methods by Near-Infrared Spectroscopy"

_foods, 2023, doi:10.3390/foods12020300_

Round 1

Reviewer 1 Report

The authors evalauted the cability of NIR spectroscopy combined with chemometrics for determination of the meat quality. I find this study valuable and important

Study is well-organized, authors explained each section in the paper in detail.

Please highlight the importance of the study in the last paragraph of introduction.

Please add reference for the protein, fat and moisture analysis methods.

Please add a table to show meat related bands, ranges and band assignments.

Please explain why authors selected the spectral regions of 400-1050 nm and 900-1700 nm.

Please define and explain the importance of Rc, SEC, Rp and  SEP values, how authors determined the best prediction method.

I couldnt see the discussion section, please discuss your result with previous contributions, there should be previous contributions about quality avaluation of NIR by using NIR or IR spectrocopy

Please perform native English check.

Author Response

Thanks to the reviewer’s comments, detail answers please see the attachment.

Reviewer 2 Report

This manuscript has merit to be published but there are numerous grammatical errors and several areas that don’t make sense within. It is not new material being examined – the authors note that several studies have used NIR spectroscopy in meat science, however the present manuscript offers a comprehensive overview of NIR spectroscopy and prediction modelling for chemical composition of lamb meat, going deeper than several published studies and comparing two spectrometers. There should be more comparison between the two spectrometers used as well as their wavelength range, particularly given their cost, size and levels of precision and accuracy. 

I would recommend some review by a native English speaker or running spell/grammar check in English, and would avoid starting sentences with abbreviations or with “And”.

General comments:

L10 – remove “kinds of”

L12, L253 – you have referred to 900-1700 nm as long-wave NIR, what is the region that goes from 1700-2500 nm?

L24 – remove “the”, change “which could provides rich” to “as it provides”

L43 – this could be written a bit better as it looks like an evolutionary spectrum of NIR devices in research, it may be worth mentoning that with technological advances the devices become smaller and more affordable to regular people

L49 – change “Gauss peak type » to “Gaussian-type peak”

L57 – change “as the seriously” to “due to the”

L55 – “So the authors doubts…”? This doesn’t make sense

L70 – add “data” after spectral

L84 – remove “temperature of”

L85 – change “slaughtered” to “slaughter”

L90-113 – how many scans were taken upon each sample? And how many scans was the spectral data collected from the sensors an average of?

L103 – 5 days x 5 carcases per day x 6 primal cuts per carcase = 150 primal cut spectra. Where on each primal cut was the scan made? Primal cuts (L86 list – topside, backstrap, oyster blade, fillet, thick flank and tenderloin) are not that small that one scan would represent the entire cut.

L154 – how many data points for each spectrometer? State the values rather than “hundreds even thousands”

L168 – keep in mind the animals have been referred to as lambs throughout the manuscript

L200-201 – were first and second derivatives considered for pre-processing?

L241 – what is “response regular”?

L244-248 – it’s best if the peaks and troughs are listed to what they respectively correspond to (bonds, overtones, etc) in order to develop the prediction models

L225 – the range of fat is good for prediction model development, but can this be ascribed to the different cuts of meat used? Similarly in L228 several factors that are listed are not relevant to the present study (except for breed) and should be there with reference, but cut is not listed as a factor at all? I urge you to check the paper in Animal Production Science (DOI: 10.1071/AN21069) which explores a similar topic

L234-235 – was there any spectral trimming taking place to remove machine artifacts being present in the spectra?

Table 2 – bold shows the optimal model, list in the title or footnote and whether it was chosen through Rc, Rp, SEC or SEP. In the case of two showing the same values (Fat with SNV and MSC at 900-1700 nm), why was SNV chosen over MSC? Similarly moisture with SNV and MSC at 900-1700 nm

L302 – coefficients are stronger than what? Or is R = 0.30 the threshold for “strong” designation? Based on other studies and the Rc/Rp values in the present one the “strong” determination should be much greater

L325 – were the two spectrometers compared for their predictive performance, or were they designed to be complimentary to one another as there is only a short window of overlap (900-1050 nm, with these potentially being poor spectra due to artifacts)?

L334 – when was the Rp of moisture content 0.79?

L335 – revise “as this number is almost keeping the old way”, it is confusing

L336 – when was the Rp of protein content 0.75? Is it “little decline” from the 400-1050 nm Rp value? I would suggest calling it “a slight decrease”

L339 – when was the Rp of raw spectra above 0.75? In Table 2 it is not above 0.75 for moisture or protein and is always less than the corresponding Rp with SNV pre-treatment

L348 – change “the modelling results shows” to “and constantly appeared”

L366 – “Rp rising from 0.40 to 0.90”, is this based on Table 3 because moisture content Rp is 0.72 at the lowest and 0,84 at the greatest

L373 – change “are” to “being”

Figure 3 – would suggest using different symbols for these graphs as they are difficult to see without colour or magnification; also the bottom row should be D, E, F rather than C, D, E

L399 – change “with” to “to”

L404 – change “fall into worst” to “decreased” and remove “was decrease to”

L407-409 – these sentences do not make sense “And it did not seem to increase…”

L415 – change “satisfying” to “satisfy”

L418 – change “and” to “in”

L420 – change “seems like slide”, not sure what this means, is this a virtual spectrum as listed in the figure footnote? If so label it and put “(Figure 4C)” after (JFPT)

L426 – change “sharp different range” to “sharply different”

L442 – change “near infrared spectrometer based on high cost and high resolution” to “high cost and high resolution near infrared spectrometer”

Conclusion itself is somewhat defeatist, would recommend revising lines 450-451 and provide a solution to reducing sensitivity manually, is there an industrial uptake of NIR sensors to predict chemical composition of lamb meat? If not, the conclusion is correct in saying that “it is the most common method to build prediction model with the whole spectrum”, what is the next step the present study and future studies can offer? If the larger spectrometer with thousands of spectral wavelength intervals is bulky and redundant in terms of modelling, the study needs to sell the smaller one with less robust spectral information, and potentially the use of only certain wavelengths to save time.

Author Response

(The authors gave the same response as above.)

Reviewer 3 Report

Table 2_the words quality, fat and moisture should be rearranged as Quality, Fat and Moisture.

Line 199_201: The methods should be explained with details in terms of differences.

Line 272: Patel N. et al, 2021) is should be changed as numerical.

At the conclusion section, the results can be summarized according to the specific objectives of this research.

objectives (1) to develop 67 two new methods to extract spectra characteristics of visible-near infrared spectra; (2) to 68 compare the prediction performance of calibration models with different sources of spec- 69 tral including raw spectra and four different method of spectral feature extraction; (3) to 70 establish prediction models of meat quality based on more potential and simple methods 71 to extract the spectral features.

Author Response

(The authors gave the same response as above.)

Round 2

Author Response

Thank you for your comments concerning our manuscript. Those comments are all valuable and very helpful for revising and improving our paper. We have studied comments carefully and have made the correction accurately according to your every comment, which we hope meet with approval.

Special thanks to you for your good comments and extremely careful checks, you are an professional scientist I admire a lot.
